# Effects of Long-Term Fasting on Gut Microbiota, Serum Metabolome, and Their Association in Male Adults

**DOI:** 10.3390/nu17010035

**Published:** 2024-12-26

**Authors:** Feng Wu, Yaxiu Guo, Yihua Wang, Xiukun Sui, Hailong Wang, Hongyu Zhang, Bingmu Xin, Chao Yang, Cheng Zhang, Siyu Jiang, Lina Qu, Qiang Feng, Zhongquan Dai, Chunmeng Shi, Yinghui Li

**Affiliations:** 1State Key Laboratory of Trauma and Chemical Poisoning, Third Military Medical University, Chongqing 200038, China; 2State Key Laboratory of Space Medicine, China Astronaut Research and Training Center, Beijing 100094, Chinayinghuidd@vip.sina.com (Y.L.); 3Department of Human Microbiome, School and Hospital of Stomatology, Cheeloo College of Medicine, Shandong University, Jinan 250012, China; 4Engineering Research Center of Human Circadian Rhythm and Sleep, Space Science and Technology Institute (Shenzhen), Shenzhen 518000, China; 5School of Life Science and Technology, Harbin Institute of Technology, Harbin 150080, China

**Keywords:** long-term fasting, gut microbiome, metabolome, metabolic homeostasis, gut bacteria-derived metabolites, *Ruthenibacterium lactatiformans*

## Abstract

Background: Long-term fasting demonstrates greater therapeutic potential and broader application prospects in extreme environments than intermittent fasting. Method: This pilot study of 10-day complete fasting (CF), with a small sample size of 13 volunteers, aimed to investigate the time-series impacts on gut microbiome, serum metabolome, and their interrelationships with biochemical indices. Results: The results show CF significantly affected gut microbiota diversity, composition, and interspecies interactions, characterized by an expansion of the Proteobacteria phylum (about six-fold) and a decrease in Bacteroidetes (about 50%) and Firmicutes (about 34%) populations. Notably, certain bacteria taxa exhibited complex interactions and strong correlations with serum metabolites implicated in energy and amino acid metabolism, with a particular focus on fatty acylcarnitines and tryptophan derivatives. A key focus of our study was the effect of *Ruthenibacterium lactatiformans*, which was highly increased during CF and exhibited a strong correlation with fat metabolic indicators. This bacterium was found to mitigate high-fat diet-induced obesity, glucose intolerance, dyslipidemia, and intestinal barrier dysfunction in animal experiments. These effects suggest its potential as a probiotic candidate for the amelioration of dyslipidemia and for mediating the benefits of fasting on fat metabolism. Conclusions: Our pilot study suggests that alterations in gut microbiota during CF contribute to the shift of energy metabolic substrate and the establishment of a novel homeostatic state during prolonged fasting.

## 1. Introduction

Fasting has a long history for religious, ethical, or health reasons. Accumulating evidence shows that fasting has many beneficial effects on human health and lifespan. Therefore, fasting has been a feasible and popular diet strategy for preventing and treating chronic metabolic disorders [1], cancer [2], and even as protection from the toxicity of chemotherapy [3]. However, the practical application of fasting interventions has become increasingly challenging due to the distinct and individual consequences of short-term complete fasting (CF) for up to two days in currently prevalent forms [4,5]. More information is also necessary for the therapeutic prescription of prolonged fasting [6]. Our earlier research showed that individuals could well tolerate a 10-day CF and achieved new metabolic homeostasis between days 3 and 6 of CF, which could provide evidence for the development of new fasting strategies for clinical practice or directing prescriptions [7]. However, further research is needed to better understand the mechanism of positive impacts. Gut microecology is considered a crucial environmental element that strongly contributes to optimizing energy metabolism and improving host health [8]. Different fasting regimens influence the gut microbiota, which has been shown to have various beneficial effects on almost every organ and tissue in animal and human studies [9,10]. However, most human studies only showed flimsy connections between fasting and the gut flora. It is still unclear whether and how gut bacteria mediate the metabolic effects of fasting, especially during a prolonged CF [10,11]. Here, we discussed how the gut microbiota and blood metabolites changed over the course of a 10-day CF in healthy adults.

The human gut contains an immense number of microorganisms, which can interact with the host’s signaling pathway to promote energy harvest and storage [12]. The gut microbiota communicates with the body by producing various metabolites, such as trimethylamine-N-oxide (TMAO) [13], secondary bile acids [14], short-chain fatty acids (SCFAs) [15], vitamins [16], etc. These metabolites are strongly linked to whole-body metabolism. SCFAs produced by the gut microbiota provide 5% to 25% of the host’s energy requirements [17]. Short-term fasting has an impact on the gut microbiota’s ability to produce SCFAs [18,19]. After a 5-day Bigu-style fast with an energy intake of <100 kcal/d, the gut microbiota altered significantly, especially the increased taurine-utilizing bacteria [20]. Ramadan fasting boosted the gut microbiome’s diversity and richness, as well as the number of beneficial bacteria such as *Akkermansia muciniphila*, *Butyricicoccus pullicaecorum*, and *Lachnospiraceae* [21]. *Prevotellaceae* and *Bacteroideaceae* were enhanced by time-restricted eating (TRE, 16:8 pattern) in healthy males [22]. However, the diversity and overall composition of the gut microbiome in adults with obesity did not significantly change as a result of either TRE [23] or protein restriction [24]. In patients with metabolic syndrome, calorie restriction for 8 weeks with 2 nonconsecutive fasting days per week significantly increased SCFA production and changed the relative abundance of *Acidobacteria bacterium*, *Roseburia faecis*, and *Eubacterium*, which were closely associated with reduced cardiometabolic risk factors [25]. In a trial with lower energy intake, a 7-day water-only fast reduced *Fusobacterium* and made the gut microbiome more homogenous, while juice fasting had no such effect. Interestingly, the dramatic changes in the gut microbiome persisted even after the water-only fast was stopped [26]. The Firmicutes/Bacteroidetes (F/B) ratio can vary significantly among individuals and can be influenced by diet and fasting patterns. Different populations were affected by the Buchinger fasting program (maximum: 250 kcal/d) in different ways. The F/B ratio and the abundance of dietary polysaccharide-degrading bacteria (*Lachnospiraceae* and *Ruminococcaceae*) decreased in healthy individuals, whereas host energy substrate-using bacteria like *Bacteroidetes* and *Proteobacteria* increased [27,28]. In patients with hypertensive metabolic syndrome, *Clostridial Firmicutes* increased while butyrate producers, such as *Faecalibacterium prausnitzii*, *Eubacterium rectale*, and *Coprococcus comes,* decreased. Modules involved in making propionate and breaking down mucin grew. The changes returned to baseline after 3 months, but health advantages persisted [29]. In summary, it is unclear how fasting affects the gut microbiota. More research is necessary to determine the precise benefits of fasting and clarify how the gut microbiota changes over time.

Until now, the effects of long-term CF on the human gut microbiome and metabolome have been minimally explored. In this study, we utilized metagenomic sequencing and ultra-high-performance liquid chromatography coupled with quadrupole orbitrap mass spectrometry (UHPLC-QE-MS) analysis to characterize the temporal changes in gut microbiota and serum metabolites, as well as their association throughout a 10-day water-only fasting regimen. This regimen encompassed a 3-day baseline period (BF), a 10-day CF period, a 4-day calorie restriction period (CR, the process of transiting to a normal diet), and a 5-day full recovery period (FR, signifying a complete resumption of regular diet). Our finding indicated that a 10-day CF significantly altered blood metabolites and induced shifts in the gut microbiota’s composition, diversity, and community structure. Specifically, certain gut bacteria taxa exhibited complex interactions with other microbial species and were correlated with metabolite fluctuations enriched in pathways involved in energy metabolism or protein conservation. Gut bacteria, such as the highly increased *Ruthenibacterium lactatiformans*, and metabolites, such as the elevated indolelactic acid (ILA), were implicated in modulating serum lipid metabolites and potentially exerting beneficial effects on health. Complementary animal experiments reinforced the significant role of *Ruthenibacterium lactatiformans* and ILA in mitigating obesity, glucose intolerance, and dyslipidemia induced by a high-fat diet (HFD). This establishes a bridge between basic research and practical application. These collective findings underscore the role of gut microbiota and metabolites in facilitating the transition of energy substrates and the establishment of a new homeostasis after extended fasting periods.

## 2. Materials and Methods

### 2.1. Participants

Thirteen male adults were enrolled to participate in this 10-day water-only CF under medical supervision in a controlled laboratory setting. Details have been described in our previous publications [7,30]. The participants came from 8 different provinces in China and had a mean age of 39.6 ± 7.9, a mean body weight of 72.1 ± 11.9 kg, and a mean body mass index of 24.6 ± 3.5 kg/m^2^. The exclusion criteria were: history of an eating disorder, diabetes mellitus, cancer, cardiovascular disease, metabolic diseases, tobacco or alcohol dependence, and antibiotic use within 3 months. All exclusion criteria were reviewed during a complete medical examination at Longgang Center Hospital, Shenzhen, China.

### 2.2. Study Design

The studies involving human participants were reviewed and approved by the Ethics Committee of the Space Science and Technology Institute (Shenzhen) (protocol code SISCJK201805001) and followed the ethical standards of the 1964 Declaration of Helsinki and its later amendments. Trial registration: ISRCTN17045897 (retrospectively registered on 17 May 2022). Before the experiment initiation, all participants were given detailed information about the whole experiment, including the schedule, physiological tests, and possible side effects and symptoms. Informed consent was obtained from all participants involved in the study.

The experiment lasted 22 days and consisted of four phases: a 3-day baseline period (BF), a 10-day CF period, a 4-day calorie restriction period (CR, the process of transiting to a normal diet), and a 5-day full recovery period (FR, complete return to a regular diet). During CF, participants consumed Nongfu Spring water ad libitum, which contains Ca ≥ 400, Mg ≥ 50, Na ≥ 80, K ≥ 35, and metasilicate ≥ 180 µg/100 mL, and followed a mild-intensity lifestyle program with walking, reading, browsing the internet, watching TV, etc. To avoid protein loss, they conducted a traditional Chinese mindfulness exercise (Baduanjin) for about half an hour every day. On the first day of the CR stage, a bowl of rice soup for breakfast, egg soup for lunch, and vegetable soup for dinner were provided to allow the intestines and stomach to adapt to the upcoming diet. The details are listed in Appendix A. Conventional physiological parameters, such as body weight, blood pressure, heart rate, etc., were examined every day. All participants completed the experiments without significant adverse effects from fasting.

### 2.3. Sampling

Blood samples were collected immediately after waking up (around 6:30–7:30 am) via arm venipuncture at BF (before fasting), CF3 (day 3 of CF), CF6, CF9, CR3 (day 3 of CR), and FR5 (day 5 of FR). Some samples were promptly sent to a certified hospital (Grade III Level A hospital), namely Longgang Central Hospital in Shenzhen. The other blood samples were centrifuged at 3000× *g* for 10 min and stored at −70 °C until further detection. Fecal samples were collected immediately upon natural defecation at the indicated time point. Aliquots of collected feces were immediately stored at −70 °C until use. Some fecal samples were unavailable for this study due to the influence of CF. Blood biochemical indexes (BCIs) were detected by Longgang Central Hospital, a certified Grade III Level A hospital in Shenzhen, using the clinical standard protocol, as previously described [7].

### 2.4. Metagenomics Sequencing and Analysis

The fecal samples were sent to Wekemo Tech Group Co., Ltd. (Shenzhen, China) to perform metagenomics sequencing and basic bioinformation analysis by standard methods. In brief, microbial DNA was extracted using a cetyltrimethylammonium bromide kit according to the manufacturer’s instructions. The quality and quantity of extracted DNA were assessed using the Fragment Analyzer 5400. The NEB Next UltraTM II DNA Library Prep Kit for Illumina was used to construct the DNA libraries following the manufacturer’s protocol. Then, the qualified libraries were sequenced with an Illumina NovaSeq 6000 (San Diego, CA, USA) platform using a paired-end read protocol. Raw data were filtered with Trimmomatic (illuminaclip: adapters_path: 2:30:10 slidingwindow: 4:20 minlen:50) to obtain high-quality clean reads, which were aligned to the human genome assembly hg38 using Bowtie2 software (Version 2.3.5) (parameter: --very sensitive) to filter host reads. To characterize the taxonomic composition of the metagenomic DNA sequences, Kraken2 and the self-built database of Wekemo Tech were used to annotate and classify the clean sequences of all samples with the parameter of 0.2 confidence. Then, Bracken (Bayesian Reestimation of Abundance after Classification with KrakEN) was used to predict the actual relative abundance of each sample at different phylogenetic levels (phylum, class, order, family, genus, and species). The unigene sequences were aligned against the UniRef90 using HUMAnN2 software (translated_query_coverage_threshold = 90.0, prescreen_threshold = 0.01, evalue_threshold = 1.0, translated_subject_coverage _threshold = 50.0) based on DIAMOND. The functional profiles of the non-redundant genes were obtained by annotation against the KEGG database using the DIAMOND alignment algorithm at Novogene Co., Ltd. (Beijing, China).

### 2.5. Differential Abundance Taxa Identification

For a differential analysis, the Chao1 and Shannon diversity index calculated by R packages (v4.0.2, “vegan”) were used to compare the taxa diversity differences among different groups. Beta diversity (between-sample) was assessed on the basis of Bray–Curtis dissimilarity and Jaccard distances and visualized by principal coordinate analysis (PCoA) plots. Moreover, multivariate analysis of permutations (PERMANOVA, R function Adonis (vegan, 999 permutations)) was conducted on the basis of Bray–Curtis dissimilarity to test whether the difference between the two groups was statistically significant compared to that within groups. A time-series analysis of the relative abundance of bacteria and metabolites was performed using STEM (Short Time-series Expression Miner, v1.3.13). STEM was run using the ‘log normalize data’ option, with all other default settings. A permutation-based test was used to detect differentially abundant taxa across time points. The results were considered significant at *p* < 0.05. The relative abundance of differentially abundant taxa at different time points was shown by a heatmap using the “pheatmap” package in R (ver. 4.0.2). Some data were analyzed on the Wekemo bioinformatics cloud platform (www.bioincloud.tech (accessed on 8 June 2023)) with default parameters.

### 2.6. Untargeted Metabolomics by UHPLC-MS

Frozen serum samples were submitted to Biotree Biotech Co., Ltd. (Shanghai, China) for untargeted metabolomics. A total of 100 μL of serum was mixed with 300 μL of methanol (containing 1 μg/mL internal standard) in a tube. Samples were then vortexed for 30 s and sonicated for 10 min in an ice-water bath. After incubation for 1 h at −20 °C to precipitate proteins, the mixture was centrifuged at 12,000× *g* for 15 min at 4 °C. Supernatants were collected and analyzed by a 1290 UHPLC (Waltham, MA, USA) instrument coupled with a Thermo Q Exactive Focus. The mobile phase A was 0.1% formic acid in water (positive) or 5 mmol/L ammonium acetate in water (negative). The mobile phase B was acetonitrile. The elution gradient was 0 min, 1% B; 1 min, 1% B; 8 min, 99% B; 10 min, 99% B; 10 min, 1% B; and 12 min, 1% B. The flow rate was 0.5 mL/min. The injection volume was 2 μL. The QE mass spectrometer acquired MS/MS spectra using information-dependent acquisition. Acquisition software Xcalibur 4.0.27 continuously evaluated full-scan survey MS data and triggered MS/MS spectra acquisition based on predefined criteria. ESI source conditions were sheath gas flow rate as 45 arbitrary units (Arb); aux gas flow rate as 15 Arb; capillary temperature 320 °C; full MS resolution 70,000; MS/MS resolution 17,500; collision energy 20/40/60 eV in the NCE model; and spray voltage 3.8 kV (positive) or −3.1 kV (negative), respectively. A UPLC HSS T3 column was used. Raw MS data was converted to mzML format using ProteoWizard (v3.0.20287) software and processed by XCMS (v3.2) in R. Preprocessing generated a data matrix of retention time, *m*/*z* values, and peak intensity. OSI-SMMS (v1.0, Dalian Chem Data Solution Information Technology, Dalian, China) annotated peaks using an in-house MS/MS database. Metabolite relative abundance was generated by normalizing the data to the sum. Multivariate statistical analyses, including principal component analysis (PCA) and orthogonal partial least squares-discriminant analysis (OPLS-DA), were performed using SIMCA14.1. The variable importance in the projection (VIP) of the first principal component was obtained in the OPLS-DA analysis. Metabolites with VIP values greater than 1 and *p* < 0.05 were screened as different metabolites. MetaboAnalyst 5.0 (https://www.metaboanalyst.ca/ (accessed on 21 June 2022)) performed pathway enrichment analysis of differential metabolites. MetOrigin (http://metorigin.met-bioinformatics.cn/ (accessed on 1 July 2022)) analyzed metabolite sources. Origin-based metabolic pathway enrichment analysis (MPEA) was employed for metabolic pathway analysis based on differential metabolites.

### 2.7. Quantification and Statistical Analysis

Boxes and whiskers showed quartiles with outliers as individual points in the boxplots. Subject data were presented as the mean ± SEM and scatter dots represented each subject in the bar plots. All the bar plots in this study were generated with GraphPad Prism 8 software (GraphPad Software Inc., Solana Beach, CA, USA) and differential analysis was conducted using the non-parametric Kruskal–Wallis test or ANOVA with significant criteria (* *p* < 0.05, ** *p* < 0.01, *** *p* < 0.001). The Spearman correlation coefficient was used to assess the associations between variables using OmicShare online tools (www.omicshare.com/tools/ (accessed on 28 May 2023)) with default parameters. The body weight and IGTT data were analyzed by two-way ANOVA, while the remaining data from animal experiments using ordinary one-way ANOVA methods built-in prism software.

### 2.8. Animal Experiments

Experiment process: The animal experiment was approved by the Committees of Animal Ethics and Experimental Safety of the China Astronaut Research and Training Center (ACC-IACUC-2023-013). Thirty 8-week-old male C57BL/6 mice were obtained from Beijing Vital River Laboratory Animal Technology Co., Ltd. (Beijing China). The mice were not genetically engineered, and no previous procedures were performed before the experiments. Every single mouse was considered an experimental unit. They were randomly divided into four groups: 6 mice for the control group, 8 mice for the only high-fat diet (HFD) group, 8 mice for HFD coupled with *Ruthenibacterium lactatiformans* group, and 8 mice for HFD coupled with ILA and housed in a controlled specific pathogen-free environment under a 12 h dark-light cycle (lights on from 07:00 to 19:00 h) with free access to food and water. After a one-week acclimatization period, the diet was switched to a control diet (XTCON50J) or HFD (XTHF60). Subsequently, 200 μL *Ruthenibacterium lactatiformans* strain 585-1 (2 × 10^9^ units in 0.9% saline) or ILA (20 mg/kg body weight in 0.9% saline) were administrated every other day by oral gavage. All mice were given ad libitum access to filtered water across the whole experiment. The mice were weighted every 2 weeks using a digital scale. IGTTs were conducted 2 days before sample collection. After 9 weeks, mice were anesthetized with sodium pentobarbital and then sacrificed by cervical dislocation. Whole blood was collected via retroorbital sampling and centrifuged at 3000 r/min for 15 min at 4 °C to collect serum, which was stored at −80 °C until further detection. The whole liver and epididymal fat pad were quickly separated and weighted, then partly fixed in 4% paraformaldehyde. The others were stored at −80 °C until further analysis.

To minimize potential confounders, all mice were kept in the same room during the experiments, under the same conditions of light and temperature. During the analysis of the biological samples, all involved were blinded to the identity of the samples, except those who then performed the statistical analysis.

IGTT analysis: The mice fasted overnight for 16 h and were then injected with a 20% glucose solution (2 g/kg body weight). Blood glucose was determined at a defined post-injection time point by allowing a drop of tail vein blood to fall onto a blood glucose test strip.

Measurement of lipid markers: The serum level of total CHOL, triglycerides (TG), (Nanjing Jiancheng, Nanjing, China) and low-density lipoprotein (LDL), high-density lipoprotein (HDL), D-lactic acid, and DAO (ELISA kits, Dogesce, Beijing, China) were measured according to the manufacturer’s specific instructions.

Morphology analysis of fat: The fixed epididymal fat pad was dehydrated, paraffin-coated, then sectioned and stained by hematoxylin and eosin (H&E) as previously reported [31]. Random sections of adipose tissue from each mouse were captured using a Nikon Eclipse Ti microscopy system at 20× and adipocyte size was analyzed by ImageJ (v1.53).

## 3. Results

### 3.1. CF Reconstructed the Composition and Structure of the Human Gut Microbiota

As previously observed, CF significantly impacted body composition, altered clinical health markers, and facilitated the establishment of new stable energy metabolic homeostasis between the 3rd and 6th day after CF initiation [7]. In brief, body weight decreased by 9.8% with 7.28 ± 1.46 kg after a 10-day CF. The overall composition and structure dynamics of the gut microbiota before, during, and after fasting were characterized through metagenomic sequencing of spontaneously collected fecal samples. This included 13 samples at 3 days before fasting (BF3) and FR5, 7 samples at CF3, 6 samples at CF9, and 12 samples at CR3, as depicted in Figure 1A. We did not obtain enough samples at CF6. A Venn diagram intuitively shows little alteration in the number of shared and unique genera (Appendix A) and species (Appendix A) at different time points. According to an analysis of gut microbiota diversity and differences, the Shannon index was significantly lower at CF9 than at other time points (Figure 1B). Congruent with the Venn diagram (Appendix A), there were no significant differences in the Chao1 indices (Appendix A). These results suggested richness remained unchanged, but evenness and diversity declined following fasting. Principal coordinate analysis (PCoA) based on Bray–Curtis dissimilarity (Figure 1C,D) and Jaccard distances (Appendix A) revealed significant alteration in the overall gut microbiota structure during the 10-day CF. There was a tendency towards structural recovery after food refeeding; however, persistent alterations remained at FR5. These findings show that the component, structure, and diversity of the human gut microbiota were considerably perturbed by 10-day CF.

CF significantly influenced gut microbiota composition at the phylum, genus, and species levels. The top ten bacteria with the highest abundance at different taxonomic levels were further analyzed. At the phylum level, the relative abundance of Proteobacteria and Verrucomicrobia significantly increased by 6.09-fold and 23.84-fold at CF3, and 6.80-fold and 25.00-fold at CF9, respectively, while Bacteroidetes and Firmicutes decreased by 52.58% and 30.07% at CF3, and 53.31% and 38.11% at CF9, respectively (Figure 1E). Meanwhile, the F/B ratio dramatically declined by 37.48% at CR3 and 17.79% at FR5 after CF (Figure 1F). At the genus level, there were noticeable elevations in the relative abundance of *Escherichia* (834.59% at CF3 and 955.66% at CF9), *Lachnoclostridium* (167.07% at CF3 and 448.58% at CF9), *Klebsiella* (197.72% at CF3 and 432.71% at CF9), *Alistipes* (54.66% at CF3), *Akkermansia* (23.84-fold at CF3 and 24.99-fold at CF9), and *Flavonifractor* (783.80% at CF3 and 668.89% at CF9) over CF, as well as discernible decreases in *Faecalibacterium* (32.20% at CF3 and 92.58% at CF9), *Roseburia* (42.21% at CF3 and 76.42% at CF9), *Parabacteroides* (11.107% at CF3), and *Bacteroides* (62.30% at CF3 and 56.71% at CF9) (Figure 1G and Appendix A). At the species level, subjects showed increases in *Escherichia coli* (834.61% at CF3 and 955.67 at CF9), *Clostridium bolteae* (124.86% at CF3 and 333.63% at CF9), *Bacteroides fragilis* (246.62% at CF3 and 440.63% at CF9), and *Klebsiella pneumonia* (178.10% at CF3 and 573.32% at CF9) after CF intervention, while decreases occurred in *Bacteroides vulgatus* (67.06% at CF3 and 65.79% at CF9), *Faecalibacterium prausnitzi* (32.20% at CF3 and 92.58% at CF9), *Lachnospiraceae bacterium GAM79* (93.08% at CF3 and 85.65% at CF9), *Bacteroides dorei* (16.65% at CF3 and 71.12% at CF9), *Bacteroides_ovatus* (88.52% at CF3 and 92.50% at CF9), and *Bacteroides thetaiotaomicron* (32.74% at CF3 and 42.61% at CF9) (Figure 1H and Appendix A).

Subsequently, a temporal pattern clustering analysis of the relative bacterial abundance over time series was performed using a short time-series expression miner (STEM). Several noteworthy profiles of species (Figure 2A) and genera (Appendix A) were identified over the experimental timeline. The count of species or genera exhibiting similar directional trends in the abundance changes is shown in the lower-left corner of each respective cluster. Based on STEM analysis, a permutation-based test was additionally implemented to identify species (Figure 2B) and genera (Appendix A) that exhibited significant abundance variations over the course of the experiment. The top 10 most abundant bacterial species (Figure 2C) and genera (Appendix A) are visually represented in ridgeline plots. Interestingly, among these, the butyric acid- and propionic acid-producing bacteria, *Flavonifractor plautii* and *Intestinimonas butyriciproducens*, showed a rising trend. Moreover, most of the significantly changed bacteria were observed to increase at CF3, while most of the markedly changed bacteria showed a decrease at CF9 (Appendix A). During the fasting period, there was a notable proliferation of certain bacteria belonging to the Proteobacteria phyla, such as *Escherichia coli*, *Shigella flexneri*, *Pseudomonas aeruginosa*, and *Pseudomonas otitidis* (Appendix A). Specifically, *Ruthenibacterium lactatiformans*, *Anaerostipes caccae*, *Parvimonas micra*, *Alistipes indistinctus*, and *Escherichia coli* exhibited a sustained and significant increase in abundance throughout the fasting period. In contrast, *Veillonella parvula* and *Bacteroides barnesiae* were observed to experience a pronounced and consistent decrease in their populations over the same time frame.

CF complicated the interspecies correlation of the human gut bacteria. To deeply analyze these alterations, a correlation network was constructed utilizing the Spearman correlation coefficients with a threshold of |r| ≥ 0.8 and *p* < 0.05 encompassing all samples across all time points (Figure 3A). Among these bacteria, *Ethanoligenens harbinense* and *Intestinimonas butyriciproducens*, related to ethanol and butyrate production, appeared to be associated with a greater number of other species. Fluctuations in the relative abundance of these species were exhibited in the correlation network over the five time points shown in Figure 3B. *Intestinimonas butyriciproducens* obviously increased during CF, followed by a decrease post-refeeding. *Ethanoligenens harbinense* showed little change during fasting. Correlation-based microbial interaction networks, adhering to the same statistical criteria, were further analyzed at each individual time point (Appendix A). The leading network topological parameters are shown in Figure 3C. The network density, number of edges, average degree, and degree of centralization were markedly higher during CF, indicative of a more robust and intricate network structure. Conversely, the betweenness centralization was clearly lower, suggesting a more evenly distributed influence among the microbial species during this period. These results suggest that microbial interaction networks were more robust and complex during CF than at the other detected time points.

### 3.2. CF Restructured the Serum Metabolome

Similar distribution characteristics between gut microbiota (Figure 1C) and serum metabolites (Figure 4A) during the CF experiment were found by the PCA based on Bray–Curtis distances dissimilarities. These serum metabolites were further clustered into three separate categories corresponding to pre-, during, and post-fasting phases using the K-means clustering method (Figure 4B). The composition of principal components during fasting (CF3, CF6) and the recovery phase (FR5) was found to be significantly divergent from the baseline, as evidenced by the OPLS-DA distribution of both positive and negative ions (Appendix A). With the extension of CF and subsequent food reintroduction, there was a clear discrimination between the metabolites at CF3 and CF6, CR3 and FR5, and BF and FR5, as determined by permutation testing of the OPLS-DA model. These results indicated that the metabolites differed before and after fasting. Furthermore, the persistence of these alterations even after the full resumption of food intake suggested that the fasting-inducing effects on the metabolic profile may have a more enduring impact.

The differential metabolites based on O2PLS-DA analysis with VIP > 1 and *p* < 0.05 by Simca-P were predominantly clustered in metabolic pathways mainly involving amino acid metabolism, ammonia recovery, glucose metabolism, fatty acid β oxidation, mitochondrial oxidation, carnitine synthesis, oxidation of branched-chain fatty acids, SFCA metabolism, etc. (Appendix A). These metabolism pathways were primarily centered around energy metabolism, the deamination of amino acids to provide substrates for gluconeogenesis, and the synthesis of carnitine for the mitochondrial shuttle. Although the amount of carnitine decreased dramatically throughout the 10-day CF, fatty acylated carnitine molecules, as well as α-Hydroxyisobutyric acid, 3-hydroxybutyric acid, and creatine, exhibited a considerable increase (Appendix A). The organism’s metabolic pattern was changed by CF, as evidenced by the noticeable changes in metabolite components and enriched functional clusters before, during, and after CF.

The serum metabolites in positive and negative ion modes (ESI+/ESI−) were separately analyzed by STEM (Figure 4C). Most of the metabolites showed significant increasing or decreasing trends during the prolonged CF. Notably, there was a plateau period during the 6th–9th days of fasting. The serum metabolites (ESI+ and ESI−) with increasing trends were further analyzed by metabolite set enrichment analysis (MSEA) (Figure 4D). According to the MSEA results, beta-oxidation of very long-chain fatty acids, spermidine and spermine biosynthesis, pantothenate and CoA biosynthesis, betaine metabolism, and oxidation of branched-chain fatty acid pathways were enriched in accordance with the promoted fatty acid oxidation and amino acid utilization during the fasted period. The metabolic pathways enriched by differential metabolites at each time point during CF were further analyzed by MESA (Figure 4E,G,H). Twelve commonly enriched metabolic pathways throughout the fasting period were closely related to fatty acids, carbohydrates, and amino acids (Figure 4E,F). For instance, fatty acid-associated pathways, including linoleic acid metabolism, fatty acid biosynthesis, and β oxidation, were mainly enriched during fasting, consistent with the switch of energy support from glucose to fatty acid, gluconeogenesis, and glycolysis. Moreover, glucose-alanine, urea, and ammonia recycling were mainly enriched in CF6 and CF9, particularly with regard to decreased citrulline (Appendix A), indicating that nitrogen metabolism was vital in the later phase of fasting. The relative abundance of five amino acids, including four essential amino acids (EAA), significantly increased, and six amino acids, including 1 EAA (tryptophan), markedly declined during CF (Appendix A). The concentration of urea increased during fasting but did not reach statistical significance [7]. Serum and uric creatinine only presented a mild increase, whereas serum urea remained unchanged (Appendix A).

### 3.3. The Relationship Between the Metabolites and Gut Microbiota During 10-Day CF

To ascertain global relationships between the gut microbiome and serum metabolism, we conducted Spearman correlation analysis using all samples across all time points based on (|r| > 0.8 and *p* < 0.05). As shown in Figure 5A, the dominant bacterium *Escherichia coli* and *Flavonifractor plautii* were located at the network’s core with more complex correlations and linked metabolites. Fluctuations in the relative abundance of species and metabolites in the correlation network over time courses were exhibited in Figure 5B. These two bacteria increased on the 3rd and 9th day of CF, then gradually recovered after refeeding. In addition, the well-known beneficial bacteria *Akkermansia muciniphila* and *Streptococcus thermophilus* were also highly enriched, suggesting that changes in these bacterial species could impact most metabolites. However, the enriched *Veillonella parvula*, an asaccharolytic anaerobic microbe that derives energy from organic acids, decreased on the 3rd and 9th day of CF.

Subsequently, the source of the differential metabolites was analyzed utilizing MetOrigin [32]. These metabolites were primarily classified into four groups: 4 host (human)-specific metabolites, 33 bacterial metabolites, 88 bacteria-host co-metabolites, and others (33 drugs, 144 food, 1 environment, and 211 unknown) (Figure 5C). Metabolic pathway analysis based on differential metabolites was performed using origin-based MPEA. Linoleic acid metabolism and steroid hormone biosynthesis were host-specific, according to origin-based function analysis. Microbiota-specific metabolites were enriched in phenylalanine metabolism, D-alanine metabolism, and furfural degradation (*p* < 0.05). Twenty-eight highly enriched pathways (Appendix A) were shared by co-metabolites, including ten pathways related to amino acid metabolism, seven to carbohydrate metabolism, and seven to lipid metabolism.

### 3.4. Gut Bacteria Were Associated with the Lipid Metabolism and Tryptophan During CF

To further explore the relationship between gut microbiota species and serum metabolites, Spearman correlation analysis between differential serum metabolites and species was performed (Appendix A). *Ruthenibacterium lactatiformans*, *Parvimonas micra*, *Escherichia coli*, *Flavonifractor plautii*, and *Veillonella parvula* were the top five species with more interaction links, according to a count analysis with |r| > 0.6 and *p* < 0.05 (Figure 6A). Interestingly, *Ruthenibacterium lactatiformans* was found to have a high correlation with fatty acylcarnitines, including hexadec-2-enoylcarnitine, vaccenylcarnitine, L-palmitoylcarnitine, L-acetylcarnitine, and 5-tetradecenoylcarnitine (|r| > 0.7, Figure 6B–G), which markedly increased during fasting (Appendix A). The details are listed in Appendix A with |r| > 0.6. So are *Escherichia coli*, *Parvimonasmicra*, *Flavonifractor plautii*, and *Alistipesindistinctus* with 5-tetradecenoylcarnitine, hexadec-2-enoylcarnitine, and Vaccenyl-carnitine (|r| > 0.7, Appendix A). These results imply that gut bacteria might affect fatty acid transport during prolonged CF. Next, we analyzed the correlation between gut microbiota and previously described biochemical indexes (BCIs) associated with lipid metabolism [7] (Appendix A). The top five species with significant interactions with BCIs were *Ruthenibacterium lactatiformans*, *Flavonifractor plautii*, *Anaerostipes caccae*, *Eisenbergiella tayi*, and *Parvimonas micra* with |r| > 0.6 and *p* < 0.05 (Figure 6H), and the top 5 BCIs were uric acid (UA), direct bilirubin (DBiL), low-density lipoprotein cholesterol (LDL-C), total bilirubin (TBiL), and total cholesterol (CHOL) (Figure 6I, Appendix A). Interestingly, there was a strong positive correlation between LDL-C and CHOL with the increased *Intestinimonas butyriciproducens*, which interacted with more gut bacteria during fasting (Figure 6J,K). The correlation between gut microbiota and BCIs during fasting was analyzed using redundancy analysis (RDA). There was a clear link between increased LDL-C, CHOL, apolipoprotein B (ApoB), lipoprotein a (Pa), leptin (LEP), and fasting intervention. While the decreased HDL-C, triglyceride (TG), apolipoprotein A1 (ApoA1), and glucose (GLU) also presented a prominent and significant negative correlation with the fasting intervention (Figure 6L). *Escherichia coli*, *Eisenbergiella tayi*, *Ruthenibacterium lactatiformans*, and *Flavonifractor plautii* presented high correction with increased lipid metabolism indexes (Figure 6M). These data suggest that prolonged CF may affect fatty acid oxidation and CHOL metabolism by influencing gut microbiota such as *Ruthenibacterium lactatiformans*.

As described above, fasting decreased NAA tryptophan (Appendix A). We also assessed the alteration of its derivative metabolites and their relationship with gut microbiota. Regarding the direct transformation productions of tryptophan, serotonin was markedly increased by ELISA detection, while the relative abundance of kynurenine significantly declined in the metabolome on CF9 (Figure 7A,B). The gut-derived metabolite ILA showed a continuous and considerable elevation during prolonged CF and was most highly positively corrected (r = 0.7332) with *Ruthenibacterium lactatiformans* (Figure 7C,D), as was indoline (Figure 7E,F). However, indoleacetic acid and indoxyl sulfate presented almost no notable change, and indole-3-propionic acid showed a significant decline during prolonged CF (Figure 7G–I). From the gene analysis of the microbiome, aromatic amino acid aminotransferase and tryptophanase significantly increased during 10-day fasting (Figure 7J,K), which could catalyze tryptophan toward ILA and indole, respectively. There were no changes observed in the other detected tryptophan decarboxylases and indolepyruvate decarboxylase. These results suggested that gut bacteria such as *Ruthenibacterium lactatiformans* affected serum lipid metabolites, biochemical parameters, and tryptophan metabolism during prolonged CF.

### 3.5. Effects of Ruthenibacterium lactatiformans and ILA on the Fat Metabolism Inducing by HFD

To explore the potential benefits of *Ruthenibacterium lactatiformans* and ILA on fat metabolism and obesity (which were highly elevated by about 19.71-fold and 2.04-fold during CF) and the strong correlation with fat metabolic indicators, mice were fed an HFD or co-administrated *Ruthenibacterium lactatiformans* or ILA alternately every other day for 9 weeks. Compared with the HFD group, the groups receiving HFD along with alternating administration of ILA or *Ruthenibacterium lactatiformans* showed significantly less body weight gain (Figure 8A,B). Administration of *Ruthenibacterium lactatiformans* markedly delayed the substantial body weight gain induced by HFD feeding. Similarly, epididymal fat pad and liver weights were notably lower in the HFD + *Ruthenibacterium lactatiformans* group (Figure 8C,D), as corroborated by reduced mean adipocyte size from the epididymal fat (Figure 8E,F). Additionally, *Ruthenibacterium lactatiformans* prevented HFD-induced dyslipidemia, as shown by lower serum levels of CHOL (Figure 8G), triglycerides (Figure 8H), and low-density lipoprotein (Figure 8I), and greater high-density lipoprotein (Figure 8J) than the HFD-only group. Moreover, both *Ruthenibacterium lactatiformans* and ILA mitigated HFD-induced glucose intolerance. An intraperitoneal glucose tolerance test (IGTT) revealed notably lower blood glucose concentration at 45, 60, 75, and 90 min post glucose injection with *Ruthenibacterium lactatiformans* treatment and at 75 and 90 min with ILA treatment (Figure 8K). HFD also induced an increased intestinal permeability as indicated by elevated serum D-lactate and diamine oxidase (DAO) levels, which was significantly alleviated after *Ruthenibacterium lactatiformans* or ILA administration (Figure 8L,M).

## 4. Discussion

Experiments and empirical evidence from yeast to humans have demonstrated that partial or complete fasting can improve health, postpone the onset of diseases, and slow the aging process. Metabolites and gut flora are important mediators for health improvement, particularly in overweight, type 2 diabetes, and inflammation disorders. Systematic data collection coupled with rigorous and confounder analysis may further solidify our insights into the benefits of fasting [9]. To date, insights into the compositional and functional dynamics of gut microbiota during prolonged CF have been limited, as the majority of scientific studies focused on intermittent fasting regimens typically lasting 16 to 48 h of recurring CF [33]. Reports showed that juice fasting [26] and TRE [23] had relatively limited effects on gut microbiota diversity or overall composition. However, fasting regimens involving very low daily energy intake, such as Buchinger fasting or Bigu-style fasting, have been observed to induce some changes in the abundance of certain microbial taxa [20,28], indicating a variability in the effects of the fasting regimen on gut microbiota. In the present study, we conducted a longitudinal assessment of fecal microbiota and serum metabolome from 13 healthy subjects using metagenomic sequencing and metabolomic profiling at multiple time points: before, during, and after a 10-day CF period. A systematic correlation analysis was performed to explore interactive regulation during and after long-term fasting.

The architecture and diversity of the gut microbiome are intricately associated with dietary habits. Fasting has been shown to exert strong impacts on the composition and structure of the gut microbiota, as evidenced by the assessment of alpha and beta diversity and multilevel hierarchical analysis of microbial populations. PCoA analysis showed that intestinal flora changes can partially revert to a pre-fasting state after reintroducing food for the same CF duration. These findings suggested that fasting can reconstruct intestinal microecology, consistent with previous reports [9,26], which could explain the more prolonged health effects of fasting. Variations in diversity reported in previous long-term fasting experiments can be attributed to methodological differences and individual variability [9]. In the present study, a 10-day CF regimen reduced the Shannon index, an indicator of diversity, without affecting Chao1 richness indices. This contrasts with other fasting modalities, such as Buchinger fasting, which did not alter alpha diversity measures, including Chao1 and Shannon indices [28,29]. During a 5-day Bigu-style fast, an increase in Shannon’s diversity and a concomitant decrease in Simpson’s index was observed, with PCA indicating similar community distribution [20]. A 7-day water-only fasting study reported effects on alpha diversity, yet highlighted the presence of individual heterogeneity in the response [26]. Our data indicated that a 10-day CF decreases microbial evenness and overall diversity without impacting richness.

The facultative anaerobic phylum Proteobacteria (mainly in *Escherichia*) significantly increased during CF, coherent with previous long-term fasting trials in humans [20,27,28], as well as in other species such as crucian carp [34], and king penguins [35]. As the most dominant phylum and front-line responder within the gut microbiota, Proteobacteria is considered a microbial signature of gut microbiota dysbiosis because of its sensitive response to nutritional and environmental factors [36]. While certain species within Proteobacteria can be opportunistic pathogens, others play essential roles in processes such as nitrogen fixation, contributing to health. In the complete blood count, most of the indexes did not show significant alterations at the six examined time points, except for a marked increase in MONO at CF3 and MONO% at CF9, but still within the normal range [7]. These data indicate no inflammation promotion occurred. For instance, *E. coli Nissle 1917* is a well-known probiotic that promotes intestinal homeostasis [37]. Reports suggested individuals on low-nitrogen diets did not exhibit nitrogen deficiency disorders, potentially due to nitrogen fixation by intestinal bacteria [38,39]. Preliminary data from our study hinted at alterations in the NifH gene, a key player in nitrogen fixation, during fasting; however the specific effects of nitrogen fixation required further validation. The importance of gas intake during Bigu-style fasting could be well explained if the role of nitrogen fixation were confirmed. The role of Proteobacteria during fasting, including specific differential bacterial species, remains to be further explored. Meanwhile, obligate anaerobic Bacteroidetes and Firmicutes obviously decreased during CF, with the F/B ratio persisting at a lower level even after food reintroduction. Similar to Proteobacteria, CF or intermittent fasting has been widely reported to significantly decrease the abundance of Firmicutes. The response of Bacteroidetes to fasting was variable, with reports indicating an increase, stabilization, or decrease in abundance. This variability may contribute to differences in the amount of energy intake during fasting [28,40,41,42]. Divergences in experimental designs, such as fasting duration and frequency, as well as individual-specific factors, are probably contributors to the varied outcomes in studies.

As previous physiological analyses found, most of the function biochemical biomarkers related to the important metabolic organ, the liver, did not change during fasting [7]. The present results showed fasting induces a profound restructuration of the metabolome, representing a significant metabolic adaption. Food deprivation triggered extensive metabolic reprogramming, particularly affecting glucose, fat, and amino acid pathways to meet the requirements of energy metabolism. In our pilot study, PCA analysis of both positive and negative ion metabolites revealed significant discrimination in the main metabolite composition between the fasting state (CF3, CF6) or post-fasting phase (CR3, FR5) compared to the pre-fasting state (BF). Additionally, a distinction was observed among CF groups themselves (CF3 VS. CF6) (Appendix A). The markedly altered metabolites were predominantly enriched in glucose metabolism pathways closely related to glycolysis, gluconeogenesis, the Warburg effect, and the tricarboxylic acid cycle, as well as in amino acid metabolism and nitrogen utilization (Figure 4E,G,H and Appendix A). Minimizing protein breakdown was a critical survival strategy for mammals and birds during long-term food deprivation [43,44], which has garnered considerable attention for clinical implications in the context of prolonged fasting [45,46]. Fructose supplementation has been shown to spare protein during 10-day fasting by serving as an alternative endogenous amino acid for gluconeogenesis [47]. The biomarker of muscle protein breakdown, 3-methylhistidine, notably declined on day 9 of continuous fasting, compared with previous reports [48]. Reduction in citrulline (Appendix A), along with moderate elevations in serum and urine creatinine, and unaltered serum urea (Appendix A), collectively indicated nitrogen conservation for protein-sparing during CF. Teruya reported that a 58 h CF increased the abundance of SCFAs, branched-chain amino acids, and carnitines to compensate for energy substitution through gluconeogenesis [49]. This phenomenon was particularly evident in the present study, with a notable increase in EAA, particularly those with glycogenic or ketogenic amino acids (Appendix A). Creatine, an alternative energy substrate, markedly increased. All these observations imply that the body endeavors to form and maintain a new energy metabolism homeostasis, pivoting toward gluconeogenesis and ketogenesis during prolonged CF, as previously reported [7].

The gut microbiota plays a pivotal role in establishing new stability in energy metabolisms during prolonged fasting. At the species level, the changed bacteria were closely correlated to serum metabolites during prolonged CF. Notably, *Flavonifractor plautii* and *Intestinimonas butyriciproducens*, known as butyrate and propionate-producing bacteria, implied that the levels of SCFAs were affected by long-term CF as previously reported [18,50]. Butyrate was the preferred energy substrate of the colonocytes [51], while propionate was an efficient hepatic gluconeogenic substrate [52]. The SCFA-associated metabolites, such as alpha-hydroxyisobutyric acid and 3-hydroxybutyric acid, were significantly increased during CF (Appendix A). Co-occurrence analysis revealed that these two SCFA-producing bacteria had the strongest associations with other bacteria and exhibited an upward trend during fasting. CF reconstructed the gut microbiome characterized by an expansion of the Proteobacteria phylum and a decrease in Firmicutes, thereby increasing the complexity and interaction strength with SCFA-producing bacteria. SCFAs, derived from the intestinal microbes, provide a portion of the necessary energy substrates for the body during CF. Additionally, our findings suggest that energy metabolism undergoes significant reprogramming during fasting, with lipid-related metabolic pathways being significantly changed in both the host and the gut microbiota. The gut microbiota, affected by CF, showed a significant association with the serum metabolites. The relative abundance of alcohol dehydrogenase (EC1.1.1.1), an enzyme implicated in fatty acid oxidation, was found to be elevated during CF (Appendix A). Furthermore, fatty acylcarnitines, which are indicative of mitochondrial fatty acid metabolism, were highly correlated with *Ruthenibacterium lactatiformans* (Figure 6B), an organism recently implicated in butyrate-related pathways [53] and observed to increase during CF. COR, a key mediator of lipolysis in adipose tissue [54], was found to significantly increase with prolonged CF and showed a markedly positive correlation with *Flavonifractor plautii*. Correlation analysis also indicated that *Intestinimonas butyriciproducens* was significantly positively correlated with LDL-C level, while *Ruthenibacterium lactatiformans* was highly correlated with the oxidation of fatty acids during the fasting period. Our pilot study collectively implied that gut bacteria might modulate the oxidation of fatty acids during prolonged CF, contributing to the new energy metabolism homeostasis as previously reported [7].

The gut microbiota has been implicated to play a role in host metabolic reprogramming and health improvement during energy restriction, with its derived metabolites playing a crucial role in maintaining energy homeostasis [55,56]. Gut microbiota-derived metabolites can be absorbed into the circulation and contribute to ketone bodies or glucose production, thereby supporting energy homeostasis during the periodic fasting of hibernation [57,58,59]. Recent evidence demonstrated that the gut microbiota could recycle urea nitrogen to synthesize nonessential amino acids, facilitating new protein synthesis for the host in hibernating thirteen-lined ground squirrels [60]. In humans, the gut microbiota has shown potential for nitrogen fixation, with certain species such as *Klebsiella* and *Clostridiales* strains capable of this process [39]. Intermittent fasting could promote white adipose tissue browning and enhance energy expenditure by altering gut microbiota composition [61]. Transplantation of gut microbiota from calorie-restricted mice has been shown to resist obesity induced by a high-fat diet and alleviate hepatic lipid accumulation [62]. Moreover, the shift in human gut microbiota composition could be linked to the metabolic energy switch during a 10-day Buchinger fast [28]. The gut microbiota accounted for the changes in lipids during human Bigu-style fasting [20]. In the present study, gut bacteria such as *Intestinimonas butyriciproducens*, *Flavonifractor plautii*, and *Ruthenibacterium lactatiformans*, and gut bacteria-derived metabolites such as ILA were involved in lipid metabolism under fasting. *Intestinimonas butyriciproducens*, a beneficial bacteria known for its butyrate and vitamin B12 production [63], is involved in the improvement of energy metabolism [64]. *Flavonifractor plautii* has been shown to alleviate the inflammatory responses of adipose tissue [65]. ILA, a known bacterial metabolite derived from tryptophan, influenced host metabolism through its interaction with the aryl hydrocarbon receptor (AHR) [66], which is known to play a key role in glucose, lipid, and CHOL metabolism [67]. In the present study, tryptophan was the only EAA that decreased, while ILA, indoline, and the abundance of the tryptophanase gene increased, with indoleacetic acid remaining unchanged during CF. Our data further provided detailed evidence to support the notion that fasting improved systemic metabolic functions in human health, with these improvements being partly dependent on the gut microbiota. By the way, the different microbiota were significantly correlated with diet-related feelings [30], including *Ruthenibacterium lactatiformans* with hunger and fullness, but not with the desire to eat (Appendix A). Consequently, the specific bacteria species and their mechanisms contributing to health improvement during CF still need further investigation.

The effects of fasting on blood lipids, especially CHOL, are still controversial. The heterogeneity of fasting programs may produce dissimilar results due to differences in energy deficit magnitude or metabolic background [68,69,70]. Serum CHOL has been observed to decrease after TRE [22], while both triglycerides and CHOL were shown to be significantly elevated after a 1-week CF, and then generally returned to pre-fasting levels after an extended 21-day fast. The HDL-C fraction was unchanged throughout the fasting and refeeding period [71]. In some cases, fasting increased CHOL, LDL-C, and ApoB without influencing the triacylglycerol and HDL concentration after 7-day fasting [72]. Our results showed that the lipid CHOL and LDL-C increased significantly during CF and returned to pre-fasting level after refeeding, consistent with a recent meta-analysis [68]. The primary function of LDL is to deliver cholesterol to cells throughout the body. This cholesterol is used for various purposes, including the production of cell membranes, the synthesis of hormones, and other essential biological functions. The changes in LDL-C levels during fasting are not as straightforward. The clinical significance of this small change is still a subject of discussion. CHOL was a crucial component of mammalian membranes and regulated the cell membrane’s mobility, thickness, and curvature. More and more evidence has demonstrated that CHOL could induce and regulate autophagy [73,74]. On the other hand, fasting can stimulate autophagy, release CHOL, and inhibit CHOL biogenesis [75,76]. Our experimental finding leads to the hypothesis that fasting-induced CHOL elevation may prevent excessive autophagy in the body. The increased cortisol (Appendix A) reflected the stress response to the energy and could regulate glucose levels and the use of fat, protein, and carbohydrates for energy to maintain homeostasis.

Fasting is well recognized for its capacity to significantly alter the composition and structure of the gut microbiota, which in turn can have a range of health benefits. For instance, an increase in the abundance of *Flavonifractor plautii* has been associated with reduced arterial stiffness, potentially through its role in elevating levels of cis-aconitic acid [77]. Similarly, the increased *Intestinimonas butyriciproducens* was able to convert amino acid into butyrate and acetate, which could serve not only as energy substrate but also as signaling molecules that influence human health [78]. Additionally, beneficial bacteria such as *Akkermansia muciniphila* and *Streptococcus thermophilus* were also highly enriched during the correlation assessment between gut microbiome and serum metabolism. A decrease in *Veillonella parvula*, an asaccharolytic anaerobic microbe, was associated with gut inflammation, suggesting its potential role in the gut’s health status [79]. Our investigation has identified a strong correlation between certain bacteria and serum biochemical indexes (BCI) related to lipid metabolism and tryptophan-derived molecules. Notably, *Ruthenibacterium lactatiformans*, which produced the highest number of correlation links in our pilot study, is recognized for its lactate output. This bacterium may function as an energy substrate and a signaling molecule, contributing to the regulation of overall body health [80].

*Ruthenibacterium lactatiformans* was demonstrated to offer protection against HFD-induced obesity and metabolic abnormalities. An increasing dietary fat ratio is an important contributor to the development of obesity, diabetes, fatty liver, cardiovascular disease, and other chronic metabolic diseases, and it presents an important aspect by which fasting can exert health benefits. To delve deeper into the role of *Ruthenibacterium lactatiformans* and ILA, which was closely related to lipid metabolism under fasting conditions, an obese mouse model was induced by an HFD. The results indicated that *Ruthenibacterium lactatiformans* could significantly attenuate body weight gain, glucose intolerance, and dyslipidemia induced by HFD (Figure 8). HFDs are known to induce obesity and concurrently impair intestinal barrier function, leading to systemic low-grade inflammation, including intestinal inflammation [81,82], as evidenced by the increase in intestinal permeability indicators, DAO and DLA [83]. *Ruthenibacterium lactatiformans* and ILA also significantly reduced the HFD-induced increments of DAO and DLA (Figure 8L,M). *Ruthenibacterium lactatiformans*, a lactic acid-producing bacteria [84], is associated with higher butyrate concentrations [53], both of which can substantially alleviate intestinal inflammation [85]. Fasting has also been recognized for its capacity to ameliorate intestinal inflammation by enhancing gut barrier integrity [86]. HFDs have been observed to decrease the *Ruthenibacterium lactatiformans* level in females [87]. Taken together, these findings suggest that *Ruthenibacterium lactatiformans* may play a mediating role in the fasting-induced benefits of fat metabolism. If it were demonstrated through a large-scale, popular study, *Ruthenibacterium lactatiformans* might be considered a probiotic candidate for ameliorating fatty chronic metabolic diseases.

Limitation of the study: This pilot study has demonstrated the close relationships between altered gut bacteria with different metabolites and biochemical fat metabolic indexes. Meanwhile, the sample size was relatively small, especially the fecal sample during CF. The function of the changed and associated gut bacteria needs to be further explored, although *Ruthenibacterium lactatiformans* was demonstrated to protect against HFD-induced obesity and metabolic abnormalities.

## 5. Conclusions

In summary, our pilot study meticulously characterized the temporal dynamics of the gut microbiome and serum metabolome during a 10-day CF in male subjects. The fasting-induced reconfiguration of the gut microbiome and metabolome revealed intricate and robust interrelationships, highlighting the complexity of metabolic adjustment during prolonged CF. The observed alterations of gut microbiota preferred to enhance SCFA production, fat, and tryptophan metabolisms. These alterations are posited to have facilitated a pivotal shift in energy metabolic substrates and the formation of a new homeostatic balance within energy metabolism, combined with a reconstructed metabolome. Our results lay the groundwork for future research aimed at devising therapeutic strategies and intervention methods. These may leverage the potential of long-term CF and modulation of the gut microbiota to address specific metabolic disorders, offering a promising avenue for personalized medicine and nutritional intervention. If proven effective in a broad study, *Ruthenibacterium lactatiformans* could be a probiotic for treating fatty metabolic diseases.

## Figures and Tables

**Figure 1 nutrients-17-00035-f001:**
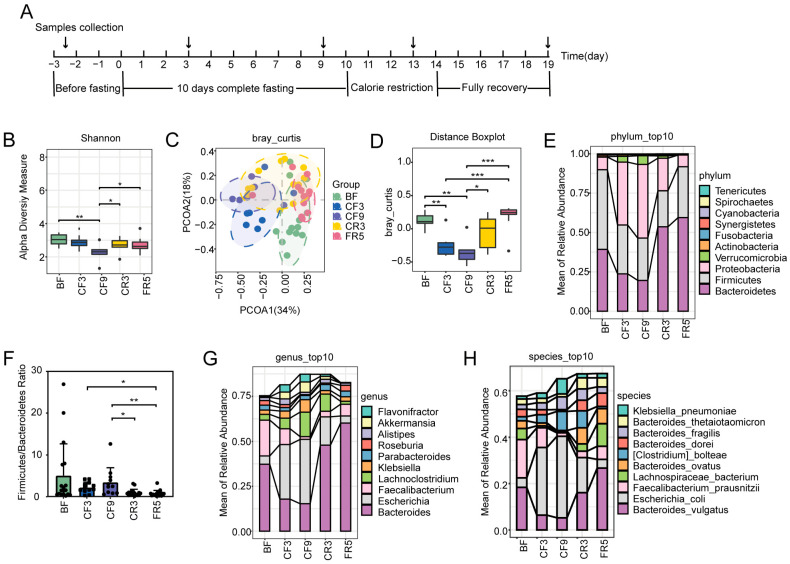
The impact of 10-day complete fasting on human gut microbiota’s diversity, difference and composition. (**A**) A schematic of the study design and the time points of sample collection. (**B**) Comparison of alpha-diversity based on Shannon index in the gut microbiota at different time points. (**C**) Principal coordinate analysis (PCoA) plot of the gut microbiota during complete fasting experiment based on the Bray–Curtis distances. (**D**) The distribution of Bray–Curtis distances from samples among the different courses in the complete fasting experiment based on the abundance. (**E**,**G**,**H**) The stacked bar plot showed the relative abundance of the gut microbiota at the phylum (**E**), genus (**G**), and species (**H**) levels, respectively. Each bar represents the mean of all detected samples at each time point. (**F**) The Firmicutes to Bacteroidetes ratio at each time point. Boxes and whiskers showed quartiles with outliers as individual points. * *p* < 0.05, ** *p* < 0.01, *** *p* < 0.001; Significant difference (*p* < 0.05) determined by Wilcox test. (**B**,**D**,**F**), PERMANOVA test. (**C**) BF: before fasting; CF3^:^ 3rd day of complete fasting; CF9^:^ 9th day of complete fasting; CR3^:^ 3rd day of calorie restriction; FR5^:^ 5th day of full recovery. Sample number is 13 at BF3 and FR5, 7 at CF3, 6 at CF9, and 12 at CR3.

**Figure 2 nutrients-17-00035-f002:**
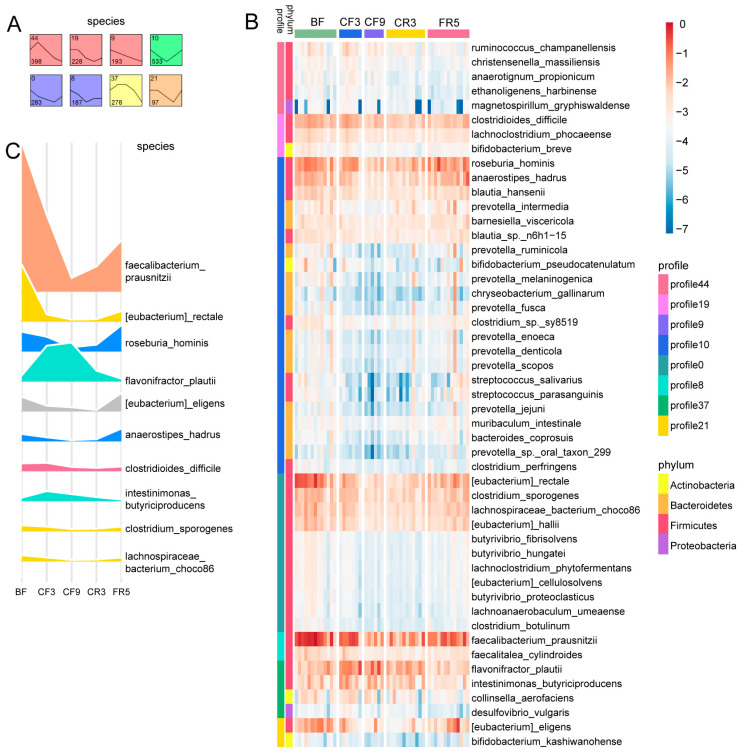
Ten-day complete fasting induced the different change profiles of human gut species. (**A**) Clustered profiles of changed species across the complete fasting times inferred by STEM analysis. Statistically significant profiles (*p* < 0.05) are represented in color. Similar colors represented the same type of change profile. The upper left number is the profile ID, and the lower left number presents the species count in each box. (**B**) Heatmap of the relative abundance of the species with significant difference (based on the Permutation test) using log10(X + min(X [X! = 0]) (X: the relative abundance of the species)) by R with colors gradually changing from blue to red, corresponding to low and high relative abundance, respectively, and trend (based on the STEM analysis) (*p* < 0.05) during the complete fasting experiment. (**C**) The ridgeline plot shows the top 10 most abundant species in the heatmap. BF: before fasting; CF3: 3rd day of complete fasting; CF9: 9th day of complete fasting; CR3: 3rd day of calorie restriction; FR5: 5th day of full recovery.

**Figure 3 nutrients-17-00035-f003:**
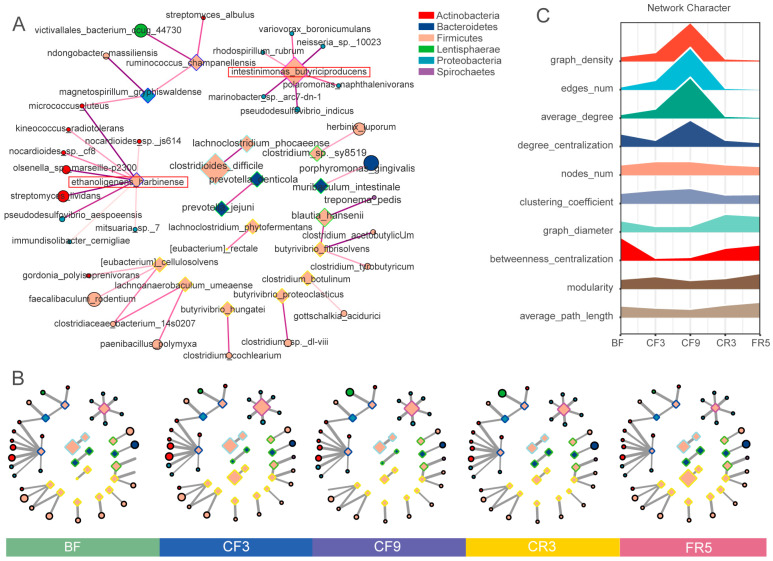
Ten-day CF impacted the correlation of the human gut microbiota. (**A**) Network analysis of the interactions between the differential species based on the Spearman correlation coefficients (|r| ≥ 0.8 and *p* < 0.05). The fill color of the circles and diamonds represented the corresponding phylum. *Ethanoligenens harbinense* and *Intestinimonas butyriciproducens* connected with more species. (**B**) The relative abundance fluctuations of the species in the correlation network over the five time points. The node size positively correlated with its relative abundance. (**C**) The ridgeline plot shows the network topological parameters. BF: before fasting; CF3: 3rd day of complete fasting; CF9: 9th day of complete fasting; CR3: 3rd day of calorie restriction; FR5: 5th day of full recovery.

**Figure 4 nutrients-17-00035-f004:**
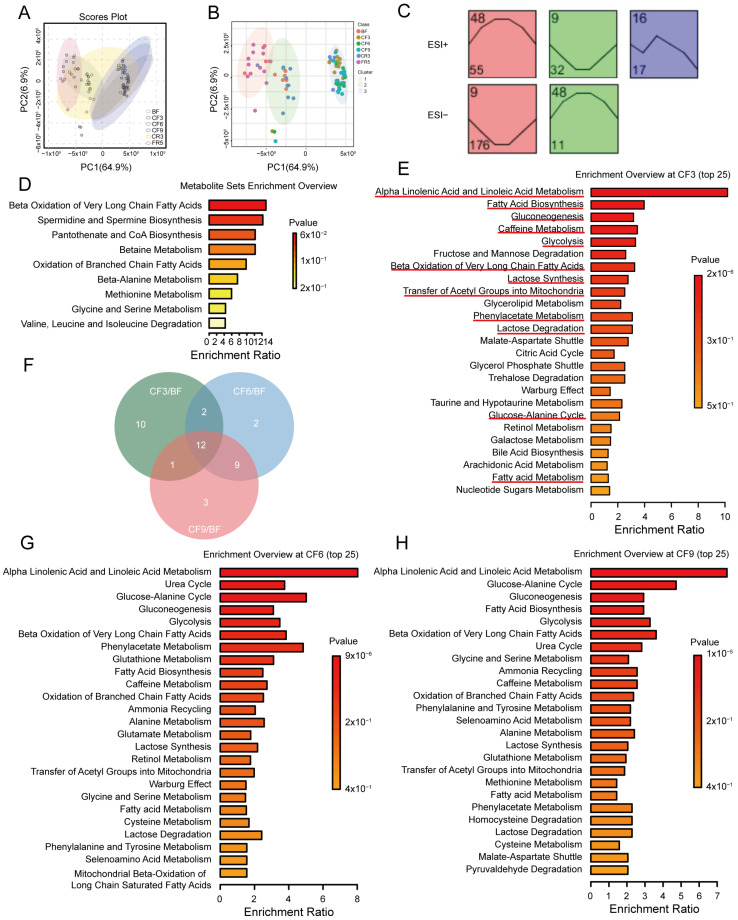
Ten-day complete fasting restructured serum metabolome. (**A**) The principal component analysis (PCA) plot of serum metabolites during the CF experiments based on Bray–Curtis distances. (**B**) Clustered results in the serum metabolites with K-Means. (**C**) Significant change profiles (*p* < 0.05) of serum metabolites in ESI+ and ESI− modes across the different time points by STEM analysis. The upper left number was the profile ID, and the lower left number presented the metabolite count in each box. (**D**) Metabolite set enrichment analysis (MSEA) of the significantly enriched and affected metabolic pathways for the serum metabolites (both ESI+ and ESI−) with an increasing trend. (**E**,**G**,**H**) The top 25 enriched metabolic pathways of differential metabolites at CF3 (**E**), CF6 (**G**), and CF9 (**H**) using the MetaboAnalyst metabolic pathway analysis tool. (**F**) Venn diagram of the number of enriched metabolic pathways among CF3, CF6, and CF9. The 12 common pathways were highlighted in Subfigure (**E**) with a red underline. ESI+: positive electrospray ionization; ESI−: negative electrospray ionization; BF: before fasting; CF3: 3rd day of complete fasting; CF9: 9th day of complete fasting; CR3: 3rd day of calorie restriction; FR5: 5th day of full recovery.

**Figure 5 nutrients-17-00035-f005:**
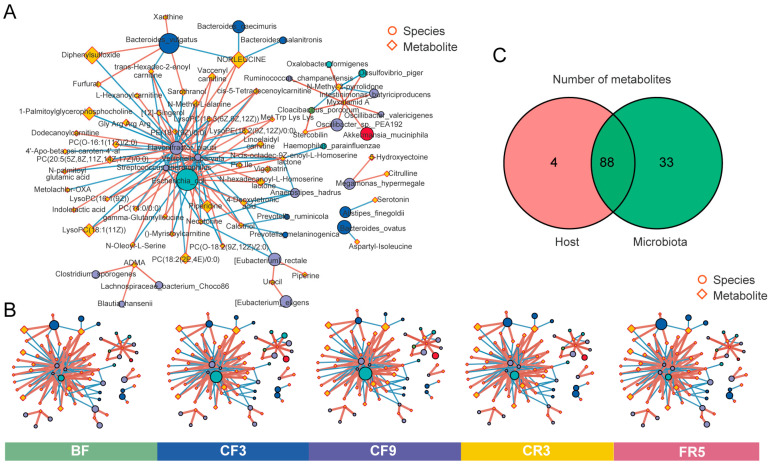
The relationship between differential metabolites and gut microbiota during a 10-day complete fast. (**A**) Network diagram of the correlation between gut microbiota and serum metabolites. The circle and diamond represented the bacteria species and metabolites, respectively. The node size positively correlates with relative abundance. The fill color of the circle represented its corresponding phylum. The line thickness was proportional to the absolute value of the correlation coefficient (|r| > 0.8 and *p* < 0.05). The red line means a positive correlation, and the blue line means a negative correlation. (**B**) The relative abundance fluctuations of gut microbiota and serum metabolites in the correlation network over the five time points. The node size positively correlates with relative abundance. (**C**) Venn diagram of the metabolite number from host or bacteria. BF: before fasting; CF3: 3rd day of complete fasting; CF9: 9th day of complete fasting; CR3: 3rd day of calorie restriction; FR5: 5th day of full recovery.

**Figure 6 nutrients-17-00035-f006:**
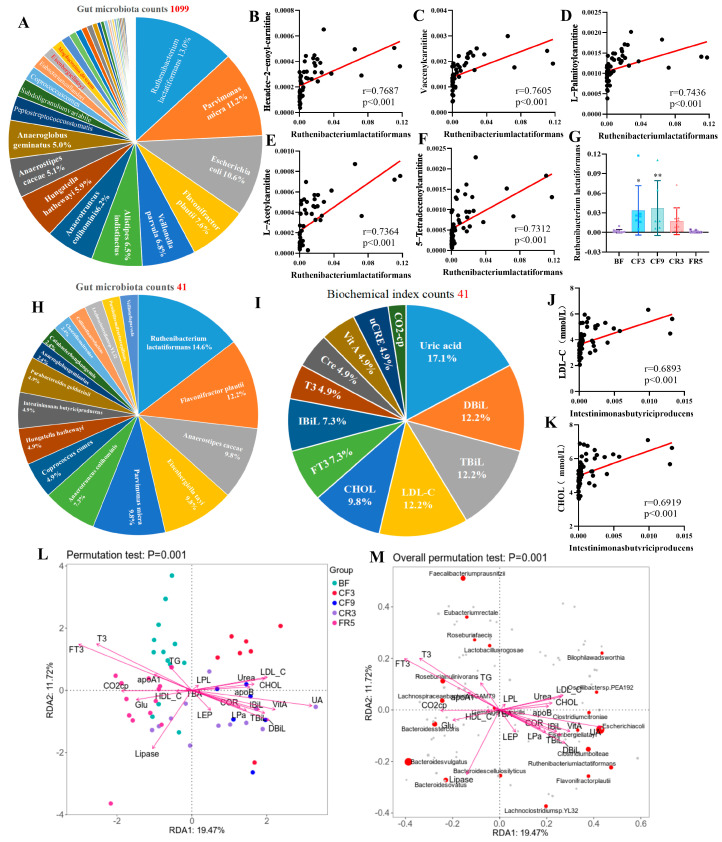
Correlation analysis between serum differential metabolites or fat metabolism relative biochemical indexes (BCIs) and differential gut microbiota during the 10-day complete fasting. (**A**) The percent of gut microbiota counts with |r| > 0.6 and *p* < 0.05 with serum metabolites. The correlation diagram between *Ruthenibacterium lactatiformans* and fat acylcarnitine (**B**) Hexadec-2-enoylcarnitine, (**C**) vaccenylcarnitine, (**D**) L-palmitoylcarnitine, (**E**) L-acetylcarnitine, and (**F**) 5-tetradecenoylcarnitine) with |r| > 0.7 and its relative abundance changes of *Ruthenibacterium lactatiformans* (**G**). * *p* < 0.05, ** *p* < 0.01, VS. BF, n = 13. (**H**) The percent of gut microbiota counts with |r| > 0.6 and *p* < 0.05 with fat metabolism relative to BCIs. (**I**) The percent of fat metabolism relative BCIs counts with |r| > 0.6 and *p* < 0.05 with gut microbiota. (**J**,**K**) Correlation diagram of low-density lipoprotein cholesterol (LDL-C) and total cholesterol (CHOL) with *Intestinimonas butyriciproducens*. (**L**,**M**) The redundancy analysis (RDA) between differential gut microbiota and BCIs during fasting. BF: before fasting; CF3: 3rd day of complete fasting; CF9: 9th day of complete fasting; CR3: 3rd day of calorie restriction; FR5: 5th day of full recovery.

**Figure 7 nutrients-17-00035-f007:**
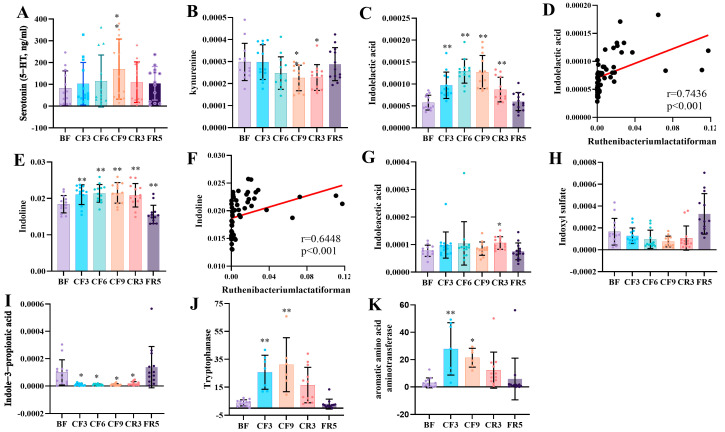
Serum level changes of tryptophan derivative metabolites and enzymes during 10-day complete fasting. (**A**) Serotonin detected by ELISA; (**B**,**C**,**E**,**G**–**I**) The relative abundance changes original from metabolome. (**D**,**F**) The correction between indolelactic acid, indoline, and *Ruthenibacterium lactatiformans*. (**J**,**K**) The relative abundance of Tryptophan metabolic enzymes original from metagenome sequencing. * *p* < 0.05, ** *p* < 0.01, VS. BF. BF: before fasting (n = 13); CF3: 3rd day of complete fasting (n = 13 for serum and n = 7 for fecal); CF6: 6th day of complete fasting (n = 13); CF9: 9th day of complete fasting (n = 13 for serum and n = 6 for fecal); CR3: 3rd day of calorie restriction (n = 13 for serum and n = 12 for fecal); FR5: 5th day of full recovery (n = 13).

**Figure 8 nutrients-17-00035-f008:**
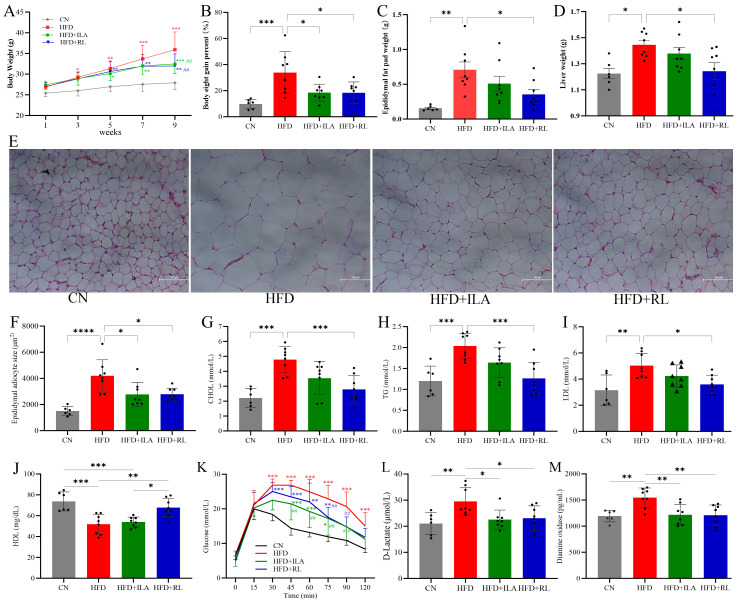
The protective effects of *Ruthenibacterium lactatiformans* and ILA on HFD-induced obesity and metabolic abnormalities. Mice were fed an HFD or co-administrated *Ruthenibacterium lactatiformans* or ILA alternately every other day for 9 weeks. (**A**) body weight every week. (**B**) body weight gain percent at 9th week. The weight of epididymal fat pad (**C**) and liver (**D**). The HE staining (**E**) of epididymal fat and adipocyte size analysis (**F**). The serum concentration of CHOL (**G**), triglycerides (**H**), low-density lipoprotein (**I**), high-density lipoprotein (**J**), D-Lactate (**L**) and Diamine oxidase (**M**). (**K**) intraperitoneal glucose tolerance test. CN: with control diet, HFD: high-fat diet, ILA: indolelactic acid, RL: *Ruthenibacterium lactatiformans,* CHOL: total cholesterol, TG: triglycerides, LDL: low-density lipoprotein, HDL: high-density lipoprotein. AUC: area under curve, IGTT: intraperitoneal glucose tolerance test. ## *p* < 0.01, vs. HFD; * *p* < 0.05, ** *p* < 0.01, *** *p* < 0.001, **** *p* < 0.0001, n = 6 for CN group and n = 8 for other groups.

## Data Availability

The metagenomic data presented in this study can be found in NCBI repositories. Accession ID is listed in the key resources table. Code: PRJNA1007579. Any additional information required to reanalyze the data reported in this work is available from the lead contact upon request.

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
