# Peer review of "Effects of Long-Term Fasting on Gut Microbiota, Serum Metabolome, and Their Association in Male Adults"

_nutrients, 2024, doi:10.3390/nu17010035_

Round 1

Reviewer 1 Report

Comments and Suggestions for Authors

This pilot study provides valuable preliminary insights into the metabolic and microbial reconfigurations induced by prolonged fasting. However, significant methodological and interpretative limitations temper its broader implications. While the findings lay a promising groundwork for future research, robust clinical trials, mechanistic studies, and larger, more diverse cohorts are required to substantiate claims of therapeutic potential and to assess the broader applicability of long-term fasting. Adding the information about the body weight and body composition before and after fasting will increase the significance of the study. In conclusion, it is very interesting pilot study which requires more robust follow up study taking into consideration the sex and the age of participants.

Author Response

This pilot study provides valuable preliminary insights into the metabolic and microbial reconfigurations induced by prolonged fasting. However, significant methodological and interpretative limitations temper its broader implications. While the findings lay a promising groundwork for future research, robust clinical trials, mechanistic studies, and larger, more diverse cohorts are required to substantiate claims of therapeutic potential and to assess the broader applicability of long-term fasting. Adding the information about the body weight and body composition before and after fasting will increase the significance of the study. In conclusion, it is very interesting pilot study which requires more robust follow up study taking into consideration the sex and the age of participants.

Reply: Thank you for your constructive feedback and for recognizing the value of our pilot study in providing preliminary insights into the metabolic and microbial changes associated with prolonged fasting. We appreciate your comments and suggestions for improving the study and its implications. Here is our point-by-point response to your concerns:

Methodological and Interpretative Limitations: We acknowledge the limitations in our current study design and interpretation. As a pilot study with a small sample size and male participants, our findings indeed require validation in larger, more diverse cohorts. Our study uncover some new and valuable clues which necessitates further research. In our next experiment, female will be included. By the way, we have made every effort to provide detailed explanations for some of the phenomena observed in the revised manuscript.

Regarding as the body weight and composition changes, we have collected these data but did not include it in the initial submission due to its publication in ref7. Now we incorporate a brief information of body weight changes in the initial parts of results (Line 288-289).

Robust Clinical Trials and Mechanistic Studies: We agree that robust clinical trials and mechanistic studies are necessary to substantiate the therapeutic claims of long-term fasting. Our current study was designed to identify potential alteration and probiotic candidate that may contribute the benefit health effects of prolonged fasting and be targeted in future clinical trials. We are in the process of designing experiment to further clarify the potential biological mechanisms underlying these changes and health benefits. If given the opportunity, we will perform a larger-scale clinical trial to deeply to explore the underlying biology of the observed changes in gut microbiota and serum metabolome.

Reviewer 2 Report

Comments and Suggestions for Authors

Overview of the manuscript

The manuscript focuses on the study of gut microbiota composition and metabolomic profile after 10 days of fasting in human volunteers. The authors observed that the fasting induce microbiota diversity in composition and interspecies interactions, characterized by the expansions of Proteobacteria, Bacteroidetes and Firmicutes populations. Complex interactions were detected in serum metabolites implicated in energy and amino acid metabolism. In the present study the authors mainly focus on the population of  Ruthenibacterium lactatiformans, which increases significantly during fasting and shows a strong correlation with fat metabolic indicators. Animal experiments performed to better investigate the effect of this bacteria strain on high-fat diet-induced obesity showed an important contribute in improving dyslipidaemia

GENERAL  COMMENT

The topic treated is very current and belongs to a modern exploration on the effect of the prolonged fasting on the human health. The work is well performed, constructed with a very deep and complex analytic examination of gut microbiota modification that give to the work, despite the low number of subjects used, a high scientific value.

SPECIFIC COMMENTS

The Firmicutes to Bacteroidetes ratio is widely used in your work. I suggest emphasizing the importance of this ratio in introduction section.

Bibliography

Citation older than the year 2000 should be avoided. However, I recognize that some of them can be considered important. 

Revise the need for using the older citations. In particular, I suggest replacing the citation 49, 53, 54, with more recent ones.

Author Response

The manuscript focuses on the study of gut microbiota composition and metabolomic profile after 10 days of fasting in human volunteers. The authors observed that the fasting induce microbiota diversity in composition and interspecies interactions, characterized by the expansions of Proteobacteria, Bacteroidetes and Firmicutes populations. Complex interactions were detected in serum metabolites implicated in energy and amino acid metabolism. In the present study the authors mainly focus on the population of Ruthenibacterium lactatiformans, which increases significantly during fasting and shows a strong correlation with fat metabolic indicators. Animal experiments performed to better investigate the effect of this bacteria strain on high-fat diet-induced obesity showed an important contribute in improving dyslipidaemia

Reply: Thank you for your comments. We appreciate the focus on the study's findings regarding the impact of fasting on gut microbiota and metabolomic profiles. We confirm that our study highlights the significant increase in Ruthenibacterium lactatiformans during fasting and its correlation with fat metabolism. The animal experiments indeed support the role of this bacterium in ameliorating dyslipidaemia associated with high-fat diets. We will ensure that our manuscript clearly communicates these points.

GENERAL COMMENT

The topic treated is very current and belongs to a modern exploration on the effect of the prolonged fasting on the human health. The work is well performed, constructed with a very deep and complex analytic examination of gut microbiota modification that give to the work, despite the low number of subjects used, a high scientific value.

Reply:We are grateful for your comments. Indeed, the limited sample size and the inclusion of only male volunteers are limitations of this pilot study. We uncover some new and valuable clues for subsequent in-depth research.

SPECIFIC COMMENTS

The Firmicutes to Bacteroidetes ratio is widely used in your work. I suggest emphasizing the importance of this ratio in introduction section.

Reply: Thanks very much for your suggestion. We added a sentence about the importance of the F/B ratio in relation to diet and fasting patterns in the introduction (Line 81-82).

Bibliography

Citation older than the year 2000 should be avoided. However, I recognize that some of them can be considered important. Revise the need for using the older citations. In particular, I suggest replacing the citation 49, 53, 54, with more recent ones.

Reply: Thanks very much for your suggestion. We deleted the ref 49. The ref 53 and 54 was replaced by “Butyrate and the Intestinal Epithelium: Modulation of Proliferation and Inflammation in Homeostasis and Disease (2021)” and “Microbiota-Generated Metabolites Promote Metabolic Benefits via Gut-Brain Neural Circuits (2014)”. We also replaced ref 17 to “The role of short-chain fatty acids in the interplay between diet, gut microbiota, and host energy metabolism (2013)” and ref 44 to “Diverging metabolic programmes and behaviours during states of starvation, protein malnutrition, and cachexia (2020)”. The others (ref 48 and 39) are suitable although they are older than 2000.

Reviewer 3 Report

Comments and Suggestions for Authors

interesting manuscript on a topic which is getting a lot of attention nowadays. The study is well designed (although 13 participants is low and makes statistical analysis nearly impossible). My main objection against this manuscript in its current form is the overkill in data: it gives the impression the authors did not know which results to present (and not present) and therefore just put in everything. It is not only the number of Figures, but also the fact that most Figures consist of up to 5 or 6 subfigures which makes it a real puzzle to read and try to understand everything. I would therefore advice the authors to more carefully select those results which are really important and leave out (or partly present as supplementary) other results.  The added value of the network analyses in Figures 3 and 5 is not very clear, so one could put less emphasis on this

Author Response

Interesting manuscript on a topic which is getting a lot of attention nowadays. The study is well designed (although 13 participants is low and makes statistical analysis nearly impossible). My main objection against this manuscript in its current form is the overkill in data: it gives the impression the authors did not know which results to present (and not present) and therefore just put in everything. It is not only the number of Figures, but also the fact that most Figures consist of up to 5 or 6 subfigures which makes it a real puzzle to read and try to understand everything. I would therefore advice the authors to more carefully select those results which are really important and leave out (or partly present as supplementary) other results. The added value of the network analyses in Figures 3 and 5 is not very clear, so one could put less emphasis on this

Reply: Thank you for your high praise of our manuscript. We wholeheartedly agree with your evaluation regarding the presentation of our data in figures. Following your suggestion, we have removed Figure 3C, as it indeed proved to be somewhat redundant. These figures represent the fundamental results of our entire analysis process.We believe that the other figures still hold some reference value, so we have decided to retain them. Thanks again for you deep assessments.